# Symbolic Regression via Neural-Guided Genetic Programming Population Seeding

**T. Nathan Mundhenk**
mundhenk1@llnl.gov

**Mikel Landajuela**
landajuelala1@llnl.gov

**Ruben Glatt**
glatt1@llnl.gov

**Claudio P. Santiago**
prata@llnl.gov

**Daniel M. Faissol**
faissol1@llnl.gov

**Brenden K. Petersen**
bp@llnl.gov

Computational Engineering Division
Lawrence Livermore National Laboratory
Livermore, CA 94550

## Abstract

Symbolic regression is the process of identifying mathematical expressions that fit observed output from a black-box process. It is a discrete optimization problem generally believed to be NP-hard. Prior approaches to solving the problem include neural-guided search (e.g. using reinforcement learning) and genetic programming. In this work, we introduce a hybrid neural-guided/genetic programming approach to symbolic regression and other combinatorial optimization problems. We propose a neural-guided component used to seed the starting population of a random restart genetic programming component, gradually learning better starting populations. On a number of common benchmark tasks to recover underlying expressions from a dataset, our method recovers 65% more expressions than a recently published top-performing model using the same experimental setup. We demonstrate that running many genetic programming generations without interdependence on the neural-guided component performs better for symbolic regression than alternative formulations where the two are more strongly coupled. Finally, we introduce a new set of 22 symbolic regression benchmark problems with increased difficulty over existing benchmarks. Source code is provided at www.github.com/brendenpetersen/deep-symbolic-optimization.

## 1 Introduction

Symbolic regression involves searching the space of mathematical expressions to fit a dataset using equations which are potentially easier to interpret than, for example, neural networks. A key difference compared to polynomial or neural network-based regression is that we seek to illuminate the true underlying process that generated the data. Thus, the process of symbolic regression is analogous to how a physicist may derive a set of fundamental expressions to describe a natural process. For example, Tycho Brahe meticulously mapped the motion of planets through the sky, but it was Johannes Kepler who later created the expressions for the laws that described their motion. Given a dataset $(X, y)$, where each point has inputs $X_i \in \mathbb{R}^n$ and response $y_i \in \mathbb{R}$, symbolic regression aims to identify a function $f : \mathbb{R}^n \to \mathbb{R}$ that best fits the dataset, where the functional form of $f$ is a short closed-form mathematical expression.

The space of mathematical expressions is structurally discrete (in functional form) but continuous in parameter space (e.g. floating-point constants). The search space grows exponentially with the length of the expression, rendering symbolic regression a challenging machine learning problem. It is generally believed to be NP-hard [Lu et al., 2016]; however, no formal proof exists. Given the

35th Conference on Neural Information Processing Systems (NeurIPS 2021).

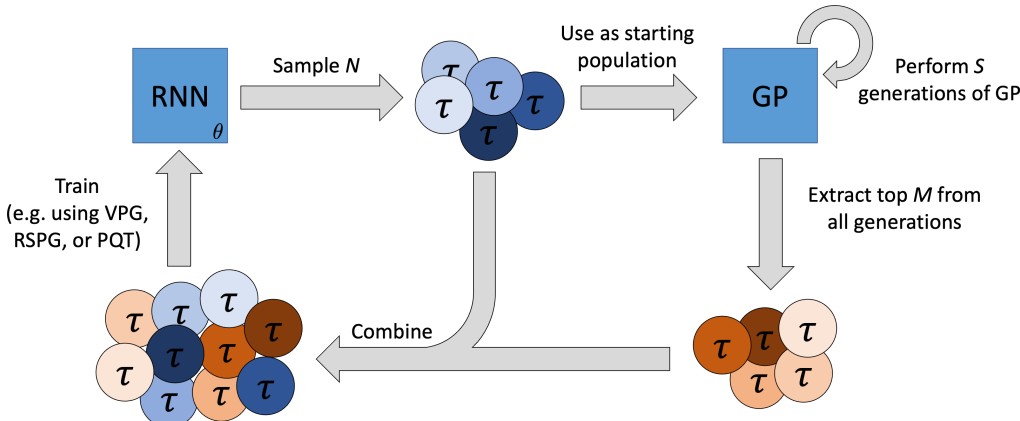

Figure 1: **Method overview.** A parameterized sequence generator (e.g. RNN) generates $N$ samples, i.e. expressions for symbolic regression. These samples are used as the starting population for a GP component. GP then runs $S$ generations. The top $M$ samples from GP are extracted, combined with the $N$ samples from the RNN, and used to train the RNN, e.g. using VPG, RSPG, or PQT. Since GP is stateless, it runs in a random restart-like fashion each time the RNN is sampled.

large, combinatorial search space, traditional approaches to symbolic regression commonly utilize evolutionary algorithms, especially *genetic programming* (GP) [Koza, 1992, Schmidt and Lipson, 2009, Fortin et al., 2012, Bäck et al., 2018]. GP-based symbolic regression operates by maintaining a population of mathematical expression "individuals" that are "evolved" using evolutionary operations like selection, crossover, and mutation. A fitness function acts to improve the population over many generations.

Neural networks can also be leveraged for symbolic regression [Kusner et al., 2017, Sahoo et al., 2018, Udrescu and Tegmark, 2020]. Recently, it has been proposed to solve symbolic regression using *neural-guided search* [Petersen et al., 2021, Landajuela et al., 2021b]. This approach works by using a recurrent neural network (RNN) to stochastically emit batches of expressions as a sequence of mathematical operators or "tokens." Expressions are evaluated for goodness of fit and a training strategy is used to improve the quality of generated formulas. Petersen et al. [2021] propose a *risk-seeking policy gradient* strategy, which filters out the lesser performers and returns an "elite set" of expressions to the RNN each step of training. Constraints can be employed to prune the search space, preventing nonsensical or extraneous statements in generated expressions. For instance, inversions (e.g. $\log(e^x)$) and nested trigonometric functions (e.g. $\sin(1 + \cos(x))$ can be avoided. The performance attained using neural-guided search outperformed GP-based approaches, including commonly used commercial software.

Genetic programming and neural-guided search are mechanistically dissimilar, yet both have proven to be effective solutions to symbolic regression. Might it be possible to combine the two approaches to leverage each of their strengths? The population of individual solutions in GP is structurally the same as a batch of samples emitted by the RNN in neural-guided search. In principle, they can formally be interfaced by allowing GP individuals to flow to the RNN training and vice versa.

For reinforcement learning-based training objectives like the one used in Petersen et al. [2021], this way of coupling creates an out-of-distribution problem on the RNN side. Standard off-policy methods, like importance sampling [Glynn and Iglehart, 1989], do not apply here since the GP distribution is intractable. Note, however, that the mechanistic interpretation of the policy gradient as maximizing the log likelihood of individuals proportional to their fitness is still valid. On the other hand, for training alternatives based on maximum likelihood estimation over a selected subset of samples, like the cross-entropy method [De Boer et al., 2005] or priority queue training [Abolafia et al., 2018], there are no assumptions about samples being "on-policy," and thus they can be applied seamlessly. In this work, we explore three different ways of training the RNN: two reinforcement learning-based training methods (without off-policy correction) and the priority queue training method.

## 2 Related Work

**Deep learning for symbolic regression.** Several recent approaches leverage deep learning for symbolic regression. AI Feynman [Udrescu and Tegmark, 2020] proposes a problem-simplification tool for symbolic regression. They use neural networks to identify simplifying properties in a dataset (e.g. multiplicative separability, translational symmetry), which they exploit to recursively define simplified sub-problems that can then be tackled using any symbolic regression algorithm. In GrammarVAE, Kusner et al. [2017] develop a generative model for discrete objects that adhere to a pre-specified grammar, then optimize them in latent space. They demonstrate this can be used for symbolic regression; however, the method struggles to exactly recover expressions, and the generated expressions are not always syntactically valid. Sahoo et al. [2018] develop a symbolic regression framework using neural networks whose activation functions are symbolic operators. While this approach enables an end-to-end differentiable system, back-propagation through activation functions like division or logarithm requires the authors to make several simplifications to the search space, ultimately precluding learning certain simple classes of expressions like $\sqrt{x}$ or $\sin(x/y)$.

**Genetic programming/policy gradient hybrids.** The overall idea of combining GP with gradient-based methods in general predates the deep learning era [Igel and Kreutz, 1999, Topchy and Punch, 2001, Zhang and Smart, 2004, Montastruc et al., 2004, Wierstra et al., 2008] and dissatisfaction with deep reinforcement learning (DRL) is not entirely new [Kurenkov, 2018]. Recently, several approaches have combined GP and reinforcement learning (RL). In these works, RL is used to alter or augment the GP process, e.g. by adjusting the probabilities of performing different genetic operations. Alternatively, GP is used to augment the creation or operation of a neural network [Such et al., 2018, Chang et al., 2018, Gangwani and Peng, 2018, Chen et al., 2019, Stanley et al., 2019, Real et al., 2019, Miikkulainen1 et al., 2019, Sehgal et al., 2019, Tian et al., 2020, Chen et al., 2020a,b].

**Sample population interchanging methods.** We identify a broad class of existing methods that can be characterized by using samples from a *sequence generator* and interchanging those samples with a GP population. The sequence generator may be any discrete distribution or generative process that creates a sequence of tokens, e.g. a recurrent neural network or transformer. Samples from the sequence generator are then treated as interchangeable with a population generated by GP. Thus, sequence generator samples can be inserted into the population of GP, and GP samples can be used to update the sequence generator. In some cases the sequence generator may not have learnable parameters.

Within this class of methods, Pourchot and Sigaud [2019] and Khadka and Tumer [2018] solve physical continuous control problems like the inverted pendulum or moon rover with a fungible neural network controller. Their approaches are similar to neuro-evolution [Stanley and Miikkulainen, 2002, Floreano et al., 2008, Lüders et al., 2017, Risi and Togelius, 2017], as both techniques are based on using deep deterministic policy gradients (DDPG) [Lillicrap et al., 2016] and utilize a synchronous shared pool of actors between GP and RL components. Khadka and Tumer [2018] introduced the ERL method that pools RL and GP actor samples into a cyclic replay buffer. This single buffer is drawn to either seed a GP step or help update the actor-critic model. Both GP and RL work synchronously, and the population is interchangeable. Pourchot and Sigaud [2019] introduce CEM-RL which solves the same family of problems in a similar way. The most notable difference is that the shared population is not fed directly into a GP, but rather it is used to update a distribution from which a population is drawn.

Most closely related to our work, Ahn et al. [2020] develop genetic expert-guided learning (GEGL). In GEGL, samples are generated using a stochastic RNN-based sequence generator and added to a maximum reward priority queue (MRPQ). A GP component then applies mutation and/or crossover on each item in the MRPQ. Notably, only one generation of evolutionary operators is applied to each sample in the MRPQ. As a consequence, there is no notion of a selection operator. In contrast, we demonstrate that the selection operator and performing multiple generations of evolution is crucial to maximize the benefits of GP. After performing one generation of GP, the resulting population is then added to a second MRPQ. Samples from the union of the GP-based MRPQ and RNN-based MRPQ are then used to train the RNN. In contrast, we find that priority queues are not necessary and may prevent sufficient exploration. As with ERL and CEM-RL, GEGL's evolutionary and training steps are one-to-one, meaning that each training step is followed by exactly one GP generation. Additionally, GP pulls from a persistent memory of sample populations, which it shares with the neural network component. This strong coupling makes the evolutionary component much more interdependent on the neural network.

# 3 Methods

Our overall algorithm comprises two main components: (1) a sequence generator (with learnable parameters) and (2) a genetic programming component. In the sections below, we first discuss preliminaries, then describe each component individually, and finally describe how the two components interact.

**Preliminaries.** Any mathematical expression $f$ can be represented by an algebraic expression tree, where internal nodes are operators (e.g. $\times, \sin$) and terminal nodes are input variables (e.g. $x$) or constants [Petersen et al., 2021]. We refer to $\tau = [\tau_1, \ldots, \tau_{|\tau|}]$ as the *pre-order traversal* of such an expression tree. Notably, there is a one-to-one correspondence between an expression tree and its pre-order traversal (see Petersen et al. [2021] for details). Each $\tau_i$ is an operator, input variable, or constant selected from a library of possible tokens, e.g. $[+, -, \times, \div, \sin, \cos, \exp, \log, x]$. A pre-order traversal $\tau$ can be instantiated into a corresponding mathematical expression $f$ and evaluated based on its fitness to a dataset. Specifically, we consider the metric normalized root-mean-square error (NRMSE), defined as follows. Given a pre-order traversal $\tau$ and a dataset of $(X, y)$ pairs of size $N$, with $X \in \mathbb{R}^n$ and $y \in \mathbb{R}$, we define $\text{NRMSE}(\tau) = \frac{1}{\sigma_y}\sqrt{\frac{1}{N}\sum_{i=1}^{N}(y_i - f(X_i))^2}$, where $f : \mathbb{R}^n \to \mathbb{R}$ is the instantiated mathematical expression represented by $\tau$, and $\sigma_y$ is the standard deviation of $y$. Finally, the fitness or reward function is given by $R(\tau) = 1/(1 + \text{NRMSE}(\tau))$. We use the terms "fitness" and "reward" interchangeably, with the former being more typical in the context of GP and the latter more common in the context of RL.

**Sequence generator.** The sequence generator is a parameterized distribution over mathematical expressions, $p(\tau|\theta)$. Typically, a model is chosen such that the likelihood of an expression is tractable with respect to parameters $\theta$, allowing backpropagation of a differentiable loss function. A common choice of model is an autoregressive RNN, in which the likelihood of the $i$th token (denoted $\tau_i$) is conditionally independent given the preceding tokens $\tau_1, \ldots, \tau_{(i-1)}$. That is, $p(\tau_i|\tau_{j\neq i}, \theta) = p(\tau_i|\tau_{j<i}, \theta)$. We follow the sequence generator used in Petersen et al. [2021]: an autoregressive RNN comprising a single-layer LSTM with 32 hidden nodes. For notational simplicity, hereafter we assume the use of an RNN as the sequence generator.

The sequence generator is typically trained using reinforcement learning or a related approach. Under this perspective, the sequence generator can be viewed as a reinforcement learning *policy*, which we seek to optimize by sampling a batch of $N$ expressions $\mathcal{T}$, evaluating each expression under a reward function $R(\tau)$, and performing gradient descent on a loss function. In this work, we explore three methods for training the RNN:

- **Vanilla policy gradient (VPG)**: Using the well-known REINFORCE rule [Williams, 1992], training is performed over the batch $\mathcal{T}$, yielding the loss function: $\mathcal{L}(\theta) = \frac{1}{|\mathcal{T}|}\sum_{\tau \in \mathcal{T}}(R(\tau) - b)\nabla_\theta \log p(\tau|\theta)$, where $b$ is a baseline term or control variate, e.g. an exponentially-weighted moving average (EWMA) of rewards.

- **Risk-seeking policy gradient (RSPG)**: Petersen et al. [2021] develop an alternative to VPG intended to optimize for best-case instead of average reward: $\mathcal{L}(\theta) = \frac{1}{\varepsilon|\mathcal{T}|}\sum_{\tau \in \mathcal{T}}(R(\tau) - \tilde{R}_\varepsilon)\nabla_\theta \log p(\tau|\theta)\mathbf{1}_{R(\tau) > \tilde{R}_\varepsilon}$, where $\varepsilon$ is a hyperparameter controlling the degree of risk-seeking and $\tilde{R}_\epsilon$ is the empirical $(1 - \varepsilon)$ quantile of the rewards of $\mathcal{T}$.

- **Priority queue training (PQT)**: Abolafia et al. [2018] introduce a non-RL approach also intended to focus on optimizing best-case performance. Samples from each batch are added to a persistent maximum reward priority queue (MRPQ), and training is performed over samples in the MRPQ using a supervised learning objective: $\mathcal{L}(\theta) = \frac{1}{k}\sum_{\tau \in \text{MRPQ}} \nabla_\theta \log p(\tau|\theta)$, where $k$ is the size of the MRPQ.

Note that our method is agnostic to the training procedure; additional procedures may be considered, for example the cross-entropy method [De Boer et al., 2005], which is closely related to PQT. In our formulation, we include, as is common, an additional term in the loss function proportional to the entropy of the distribution at each position along the sequence [Bello et al., 2016, Abolafia et al., 2018, Petersen et al., 2021, Landajuela et al., 2021b].

**Genetic programming.** Genetic programming (GP) begins with a set ("population") of expression trees ("individuals"), denoted $\mathcal{T}_{\text{GP}}$. In standard GP, these individuals are randomly generated; however, we discuss in the subsequent section how this differs in our algorithm. A single iteration or

---

**Algorithm 1** Neural-guided genetic programming population seeding

---
**input** batch/population size $N$; number of GP generations per RNN training update $S$; constraints $\Omega$; crossover probability $P_c$; mutation probability $P_m$; tournament size $k$; GP sample selection size $M$ (must be $\leq N$)
**input** Loss function $\mathcal{L}(\theta)$ for training the RNN, including corresponding hyperparameters (e.g. EWMA coefficient for VPG, risk factor for RSPG, priority queue size for PQT)
**output** Best sample $\tau^\star$
1: Initialize reward function $R : \tau \rightarrow \mathbb{R}$
2: Initialize RNN distribution over expressions, $p(\cdot | \theta, \Omega)$
3: Initialize GP operation, $\text{GP}(P_c, P_m, k, R) = \Gamma : \mathcal{T} \rightarrow \mathcal{T}$
4: $\tau^\star \leftarrow$ null
5: **while** total samples below budget **do**
6: $\quad \mathcal{T}_{\text{RNN}} \leftarrow \{\tau^{(i)} \sim p(\cdot | \theta, \Omega)\}_{i=1}^N$ $\hspace{2cm}$ ▷ Sample batch of size $N$
7: $\quad \mathcal{T}_{\text{GP}}^{(0)} \leftarrow \mathcal{T}_{\text{RNN}}$ $\hspace{2cm}$ ▷ Seed GP starting population with RNN batch
8: $\quad$ **for** $s = 1, \dots, S$ **do**
9: $\quad\quad \mathcal{T}_{\text{GP}}^{(s)} \leftarrow \Gamma\left(\mathcal{T}_{\text{GP}}^{(s-1)}\right)$ $\hspace{2cm}$ ▷ Apply GP operations
10: $\quad \mathcal{T}_{\text{train}} \leftarrow \text{Top-}M\left(\mathcal{T}_{\text{GP}}^{(0)} \cup \mathcal{T}_{\text{GP}}^{(1)} \cup \cdots \cup \mathcal{T}_{\text{GP}}^{(S)}\right)$ $\hspace{1cm}$ ▷ Filter best $M$ samples from GP
11: $\quad \mathcal{T}_{\text{train}} \leftarrow \mathcal{T}_{\text{train}} \cup \mathcal{T}_{\text{RNN}}$ $\hspace{1.5cm}$ ▷ Join RNN and best GP samples
12: $\quad \mathcal{R} \leftarrow \{R(\tau) \, \forall \, \tau \in \mathcal{T}_{\text{train}}\}$ $\hspace{2cm}$ ▷ Compute rewards
13: $\quad \theta \leftarrow \theta + \nabla_\theta \mathcal{L}(\theta)$ $\hspace{2cm}$ ▷ Train the RNN (e.g. using PQT)
14: $\quad$ **if** $\max \mathcal{R} > R(\tau^\star)$ **then** $\tau^\star \leftarrow \tau^{(\arg\max \mathcal{R})}$ $\hspace{1cm}$ ▷ Update best sample
15: **return** $\tau^\star$

---

"generation" of a GP algorithm consists of several "evolutionary operations" that directly alter the current population. A *mutation* operator introduces random variations to an individual, for example by replacing one subtree with another randomly generated subtree. A *crossover* operator involves exchanging content between two individuals, for example by swapping one random subtree of one individual with another random subtree of another individual. A *selection* operator is used to select which individuals from the current population persist onto the next population. A common selection operator is tournament selection [Koza, 1992], in which a set of $k$ candidate individuals are randomly sampled from the population, and the individual with the highest fitness is selected. In each generation of GP, each individual has a probability of undergoing mutation and a probability of undergoing crossover; selection is performed until the new generation's population has the same size as the current generation's population. GP does not have an explicit objective function as does the sequence generator, but it does tend to move the population toward higher fitness [Such et al., 2018].

For the GP component, we begin with a standard formulation from DEAP [Fortin et al., 2012] and introduce several key changes:

1. Typically only *uniform* mutation is used. Instead, we select among uniform, node replacement, insertion, or shrink mutation with equal probability.

2. We incorporate constraints from Petersen et al. [2021] (for example, constraining nested trigonometric functions, e.g. $\sin(1 + \cos(x))$). If a genetic operation would result in an individual that violates any constraint, that genetic operation is instead reverted. That is, the child expression instead becomes a copy of the parent expression. This procedure ensures that all individuals satisfy all constraints for all generations.

3. The sample population is never initialized randomly. Rather, the initial population is always seeded by samples from the RNN. (See next section for details.)

**Neural-guided genetic programming population seeding.** While both a fully RNN-based approach (i.e. deep symbolic regression [Petersen et al., 2021]) and a fully GP-based approach (i.e. GP-based symbolic regression [Koza, 1992]) involve generating "batches" or "populations" of expressions, they arrive at their expressions very differently. Namely, we observe that the RNN can generate expressions "from scratch" given only parameters $\theta$; in contrast, GP requires an extant population to operate on. More specifically, given parameters $\theta$, the RNN can be used to sample a batch of expressions, $\mathcal{T}_{\text{RNN}}$; in contrast, generation $i$ of GP begins with a population of expressions $\mathcal{T}_{\text{GP}}^{(i)}$ and application of one generation of GP produces a new, augmented population $\mathcal{T}_{\text{GP}}^{(i+1)}$.

Table 1: Recovery rate of several algorithms on the Nguyen benchmark problem set across 100 independent training runs. Results of our algorithm are obtained using PQT; slightly lower recovery rates were obtained using VPG and RSPG training (see Table 3 for comparisons).

| Benchmark | Expression | Recovery rate (%) | | | | | |
|---|---|---|---|---|---|---|---|
| | | Ours | DSR | PQT | VPG | GP | Eureqa |
| Nguyen-1 | $x^3 + x^2 + x$ | 100 | 100 | 100 | 96 | 100 | 100 |
| Nguyen-2 | $x^4 + x^3 + x^2 + x$ | 100 | 100 | 99 | 47 | 97 | 100 |
| Nguyen-3 | $x^5 + x^4 + x^3 + x^2 + x$ | 100 | 100 | 86 | 4 | 100 | 95 |
| Nguyen-4 | $x^6 + x^5 + x^4 + x^3 + x^2 + x$ | 100 | 100 | 93 | 1 | 100 | 70 |
| Nguyen-5 | $\sin(x^2)\cos(x) - 1$ | 100 | 72 | 73 | 5 | 45 | 73 |
| Nguyen-6 | $\sin(x) + \sin(x + x^2)$ | 100 | 100 | 98 | 100 | 91 | 100 |
| Nguyen-7 | $\log(x + 1) + \log(x^2 + 1)$ | 97 | 35 | 41 | 3 | 0 | 85 |
| Nguyen-8 | $\sqrt{x}$ | 100 | 96 | 21 | 5 | 5 | 0 |
| Nguyen-9 | $\sin(x) + \sin(y^2)$ | 100 | 100 | 100 | 100 | 100 | 100 |
| Nguyen-10 | $2\sin(x)\cos(y)$ | 100 | 100 | 91 | 99 | 76 | 64 |
| Nguyen-11 | $x^y$ | 100 | 100 | 100 | 100 | 7 | 100 |
| Nguyen-12 | $x^4 - x^3 + \frac{1}{2}y^2 - y$ | 0 | 0 | 0 | 0 | 0 | 0 |
| | Average | **91.4** | 83.6 | 75.2 | 46.7 | 60.1 | 73.9 |

Thus, we propose that a natural point at which to interface neural-guided search and GP is the starting population of GP, $\mathcal{T}_{\mathrm{GP}}^{(0)}$. Specifically, we propose to use the most recent batch of samples from the RNN directly as the starting population for GP: $\mathcal{T}_{\mathrm{GP}}^{(0)} = \mathcal{T}_{\mathrm{RNN}}$. From there, we can perform $S$ generations of GP, resulting in a final GP population, $\mathcal{T}_{\mathrm{GP}}^{(S)}$. Finally, we can sub-select an "elite set" of the top-performing GP samples, and include these samples in the gradient update for neural guided search (e.g. VPG, RSPG, PQT). This process constitutes one step of our algorithm, and is repeated until a maximum number of total expression evaluations is reached. Thus, GP acts as an inner optimization loop, within an outer gradient-based loop of neural-guided search.

Note that the GP process is *restarted* for each new batch of samples from the RNN; thus, the process is similar to GP with random restarts; the key difference for our proposed algorithm is that the GP starting population upon each restart *changes* as the RNN learns via an objective function. Thus, from the perspective of GP, the RNN provides increasingly better starting populations. Empirically, we found that this hybrid approach also allows for larger learning rates relative to pure neural-guided search.

We hypothesize that this hybrid approach of neural-guided search and GP will leverage the strengths of each individual approach. Whereas GP is stateless and there is no learning step, neural-guided search is stateful (via RNN parameters $\theta$) and *learns* from data via a well-defined, differentiable loss function. Whereas neural-guided search is known to easily get stuck in local optima, GP can produce large variations in the population, resulting in "fresh" samples that are essentially out-of-distribution from the RNN-induced distribution. We discuss this further in Discussion.

We provide pseudocode for our algorithm in Algorithm 1 and illustrate it in Figure 1. While this work focuses on the task of symbolic regression, our approach applies to any symbolic optimization task with a black-box reward function.

## 4   Results

We used two popular benchmark problem sets to compare our technique to other methods: Nguyen [Uy et al., 2014] and the R rationals [Krawiec and Pawlak, 2013]. A symbolic regression benchmark problem is defined by a ground truth expression, a set of sampled points from the expression, and a set of allowable tokens. Additionally, we introduce a new benchmark problem set with this work, which we call Livermore. The impetus for introducing a new benchmark problem set was that our algorithm achieves nearly perfect scores on Nguyen, so we designed a benchmark problem set with a large range of problem difficulty. Finally, we include variants Nguyen-12*, R-1*, R-2*, and R-3*, which use the same expression and set of tokens, but increase the domain over which sampled points are taken. All benchmarks are described in Appendix Table 9. Hyperparameters are shown in Appendix

Table 2: Recovery rate of several algorithms on the Nguyen, R, and Livermore benchmark problem sets across 25 independent training runs. 95% confidence intervals are obtained from the standard error between mean recovery on 37 unique benchmark problems. Recovery rates on individual benchmark problems are shown in Appendix Table 5.

| | Recovery rate (%) | | | |
| | All | Nguyen | R | Livermore |
|---|---|---|---|---|
| Ours | **74.92** | **92.33** | **33.33** | **71.09** |
| GEGL [Ahn et al., 2020] | 64.11 | 86.00 | **33.33** | 56.36 |
| Random Restart GP (i.e. GP only) | 63.57 | 88.67 | 2.67 | 58.18 |
| DSR (i.e. RNN only) [Petersen et al., 2021] | 45.19 | 83.58 | 0.00 | 30.41 |
| 95% confidence interval | ±1.54 | ±1.76 | ±2.81 | ±1.32 |

Table 11. For all algorithms, we tuned hyperparameters using Nguyen-7 and R-3⋆. None of the Livermore benchmarks were used for hyperparameter tuning.

We follow the experimental procedure of Petersen et al. [2021] unless otherwise noted. For all benchmark problems, we run each algorithm multiple times using a different random number seed. Experiments were conducted on 36 core, 2.1 GHz, Intel Xeon E5-2695 workstations. We run each benchmark on a single core, which take an average of 4.4 minutes per run on the Nguyen benchmarks. Runtimes on individual Nguyen benchmarks are shown in Appendix Table 6.

Our primary empirical performance metric is "recovery rate," defined as the fraction of independent training runs in which an algorithm discovers an expression that is *symbolically equivalent* to the ground truth expression within a maximum of 2 million candidate expressions. The ground truth expression is used to determine whether the best found candidate was correctly recovered. Table 1 shows recovery rates on each of the Nguyen benchmark problems compared with DSR, PQT, VPG, GP, and Eureqa [Schmidt and Lipson, 2009]. We note that DSR stands as the prior top performer on this set [Petersen et al., 2021]. As we can see, the recovery rate for our algorithm is 9.3% higher than the previous leader, DSR.

We next compare against a recent method called GEGL [Ahn et al., 2020] using the Nguyen, R, and Livermore benchmark problem sets. GEGL is a hybrid RNN/GP algorithm originally demonstrated for designing small molecules that fit a set of desired parameters. Many details of the open-source implementation are tied to the particular problem of molecular design (e.g. the mutation operators are specific to molecules), so we adapted their method to symbolic regression (i.e. using the same genetic operators as our method). In addition to GEGL, we compare against a "GP only" and "RNN only" version of our method, which are the most critical ablations. "GP only" does not use an RNN, and each GP population is seeded using randomly generated expressions; this is essentially GP with random restarts. "RNN only" does not use GP, essentially reducing to DSR [Petersen et al., 2021]. For these experiments, we use the variations Nguyen-12⋆, R-1⋆, R-2⋆, and R-3⋆. Table 2 shows that our method ties with GEGL on the R benchmark set and outperforms the other three methods overall. It also outperforms all other approaches on the other benchmark sets. It is interesting to note how well random restart GP does on its own, outperforming DSR and on par with GEGL. Results for individual benchmarks are shown in Appendix Table 5.

Results on additional benchmark problem sets are found in Appendix Tables 7 and 8. Finally, in Appendix Table 13 we see an additional improvement in recovery rates by repeating our experiments for the Nguyen, R, and Livermore benchmarks using the soft length prior and hierarchical entropy regularizer from Landajuela et al. [2021a], which were developed concurrently with this work.

## 5  Discussion

**Intuition.** We provide a two-sided perspective as to why we believe our hybrid method outperforms its constituent components so strongly. First, from the RNN perspective, we believe GP helps by providing "fresh" new samples that help escape local optima. NN-based discrete distributions often concentrate their probability mass on a relatively small portion of the search space, resulting in premature convergence to locally optimal solutions. In contrast, the evolutionary operations of GP generate new individuals that may fall well outside the RNN's concentrated region of the search space. Second, from the perspective of GP, the RNN helps by learning good starting populations. With a good starting set, GP can easily identify excellent solutions within a few generations. Further,

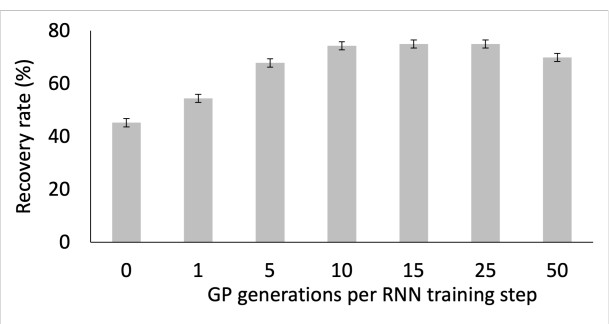

Figure 2: Recovery rate of our method from 25 independent runs on the Nguyen, R, and Livermore benchmark problem sets as a function of number of GP generations per RNN training step. Error bars represent the 95% confidence interval. The zero generations case reduces to DSR.

Table 3: Recovery rates for various ablations of our algorithm (sorted by overall performance) from 25 independent runs on the Nguyen, R, and Livermore benchmark problem sets.

| | Recovery rate (%) | | | |
| | All | Nguyen | R | Livermore |
|---|---|---|---|---|
| Trainer = PQT | **74.92** | 92.33 | **33.33** | **71.09** |
| No RL samples to train RNN (all off-policy) | 74.81 | 93.00 | 29.33 | **71.09** |
| Trainer = VPG | 74.27 | **93.67** | 28.00 | 70.00 |
| Entropy weight = 0 | 73.95 | 92.00 | 30.67 | 70.00 |
| Trainer = RSPG | 73.95 | 92.67 | 39.33 | 69.82 |
| (A): Uniform mutation only | 72.65 | 91.67 | 25.33 | 68.73 |
| (B): No inv/trig constraint | 71.35 | 91.33 | 32.00 | 65.82 |
| (A) and (B) | 68.11 | 91.33 | 21.33 | 61.82 |
| No GP samples to train RNN (all on-policy) | 66.27 | 89.00 | 24.00 | 59.64 |
| (C): Trainer learning rate = 0 | 65.95 | 90.67 | 17.33 | 59.09 |
| Random Restart (GP only) | 63.57 | 88.67 | 2.67 | 58.18 |
| (D): No RL seeds to GP | 63.24 | 89.00 | 2.67 | 57.45 |
| (C) and (D) | 63.14 | 88.67 | 2.67 | 57.45 |
| DSR (RL only) [Petersen et al., 2021] | 45.19 | 83.58 | 0.00 | 30.41 |
| 95% Confidence Interval | ±1.54 | ±1.76 | ±2.81 | ±1.32 |

the RNN may act to keep GP from moving too quickly in a suboptimal direction and may have an effect analogous to using a trust region. Each time GP restarts, it falls back to the current probability mass of the training RNN.

**GP generations per RNN training step.** An important hyperparameter in our method is $S$, the number of GP generations to perform per RNN training step. We hypothesized that $S$ be proportional to expression length. That is, the larger an expression is, the more evolutionary operations should be required to transform any arbitrary seed individual into the target expression. Our initial estimate was that 25 GP steps would be needed per each RNN training step. Figure 2 shows a post-hoc analysis of how performance varies depending on how many GP steps we do between each RNN training step. The optimal number of steps is between 10 and 25. We note that the hard limit set by our implementation is 30 tokens. However, each GP step can make many changes at a time, so we expect the number of needed GP steps to be less than the maximum possible hamming distance. As such, we can see the similarity between the optimal number of GP steps and the length of expressions.

**Ablations.** We ran several post-hoc ablations to determine the contribution of our various design choices to our method's performance. Table 3 shows the recovery rate for several ablations. We first note that the choice of RNN training procedure (i.e. VPG, RSPG, or PQT) does not make a large difference; while PQT outperforms VPG and RSPG, the difference falls within the 95% confidence interval. Interestingly, removing the entropy regularizer by setting the weight to zero made a small difference in performance. Other works have found the entropy regularizer to be extremely important [Abolafia et al., 2018, Petersen et al., 2021, Landajuela et al., 2021a]. We hypothesize that entropy is less important in our hybrid approach because the GP-produced samples provide the RNN with

Table 4: Minimum normalized mean-square error (NMSE) from 25 independent runs on three challenging benchmark problems.

| | Minimum NMSE | |
| | Ours | DSR |
| --- | --- | --- |
| Nguyen-12 | $\mathbf{1.41 \times 10^{-4}}$ | $2.01 \times 10^{-2}$ |
| R-1 | $\mathbf{3.55 \times 10^{-5}}$ | $2.74 \times 10^{-3}$ |
| R-2 | $\mathbf{2.83 \times 10^{-4}}$ | $5.31 \times 10^{-3}$ |

sufficient exploration. Using different types of mutation in the GP component with equal probability slightly improves performance; we hypothesize that this increases sample diversity. Including the constraints proposed by Petersen et al. [2021] also improves performance.

We performed several ablations designed to determine the relative importance of the RNN versus GP components. Completely removing the RNN leads to a standalone GP algorithm running with random restarts. This actually performs quite well compared to GP without random restarts. This is not surprising since random-restarts has been shown to help GP by others [Houck et al., 1996, Ghannadian et al., 1996, Fukunaga, 2010]. Similarly, we can set the RNN learning rate to zero. This ablation essentially reduces to a version of GP with random restarts, in which the starting population is based on a randomly initialized RNN; this performs similarly to GP with random restarts.

We also explore a fully "off-policy" ablation, in which only GP samples are considered in the training step (RNN samples are excluded). Notably, this does not significantly affect performance, as the GP samples (having undergone many generations of GP refinement) are generally superior to the RNN samples. Similarly, we explore a fully "on-policy" version, in which only RNN samples are considered in the training step (GP samples are excluded). In this case, there is no feedback between GP and the RNN; the only role of GP is to provide strong candidates that may be selected as the best final expression. This ablation results in a large drop in performance, suggesting that it is important that the RNN be trained on samples from the GP.

Finally, in Appendix Table 12 we show additional results when various hyperparameters from Petersen et al. [2021] or Fortin et al. [2012] are restored to their original values.

**Limitations.** The primary limiting factor to our approach is that there are still some expressions which cannot be completely recovered. While it may be a matter of finding the right set of hyperparameters, we have not as yet recovered benchmarks Nguyen-12, R-1, or R-2. While we have not yet solved three benchmarks, Table 4 shows that we still significantly outperform DSR in terms of NMSE. Other challenging benchmarks include R-3, which we recover about 1 in 25 runs, and Livermore-7 and Livermore-8, which we recover about once in every few hundred runs.

## 6 Conclusion and Future Work

We introduce a hybrid approach to solve symbolic regression and other symbolic optimization problems using neural-guided search to generate starting populations for a generic programming component running in a random restart-like fashion. The results are state-of-the-art on a series of benchmark symbolic regression tasks.

Our future research will focus on methods to mitigate or resolve the off-policy issue of policy gradient-based training methods, by either correcting the weights on the gradient computation or considering alternative optimization objectives. We will also further examine why Nguyen-12, R-1, and R-2 seem intractable and see if we can come to a solution. We also plan on doing in-depth analysis of our algorithm with noisy data.

## Acknowledgements

We thank the LLNL Hypothesis Testing team, Kate Svyatets, and Miles the cat for their helpful comments, feedback, and support. This work was performed under the auspices of the U.S. Department of Energy by Lawrence Livermore National Laboratory under contract DE-AC52-07NA27344. Lawrence Livermore National Security, LLC. LLNL-CONF-820015 and was supported by the LLNL-LDRD Program under Projects 19-DR-003 and 21-SI-001. The authors have no competing interests to report.

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
