# Appendix

Table 5: Recovery rates for individual benchmark problems.

| | Recovery rate (%) | | | |
| | DSR | Random Restart GP | GEGL | Ours |
|---|---|---|---|---|
| Nguyen-1 | 100 | 100 | 100 | 100 |
| Nguyen-2 | 100 | 100 | 100 | 100 |
| Nguyen-3 | 100 | 100 | 100 | 100 |
| Nguyen-4 | 100 | 100 | 100 | 100 |
| Nguyen-5 | 72 | 100 | 92 | 100 |
| Nguyen-6 | 100 | 100 | 100 | 100 |
| Nguyen-7 | 35 | 64 | 48 | 96 |
| Nguyen-8 | 96 | 100 | 100 | 100 |
| Nguyen-9 | 100 | 100 | 100 | 100 |
| Nguyen-10 | 100 | 100 | 92 | 100 |
| Nguyen-11 | 100 | 100 | 100 | 100 |
| Nguyen-12$^\star$ | 0 | 0 | 0 | 12 |
| **Nguyen average** | 83.58 | 88.67 | 86.00 | **92.33** |
| R-1$^\star$ | 0 | 4 | 0 | 4 |
| R-2$^\star$ | 0 | 0 | 0 | 4 |
| R-3$^\star$ | 0 | 4 | 100 | 92 |
| **R average** | 0.00 | 2.67 | **33.33** | **33.33** |
| Livermore-1 | 3 | 100 | 100 | 100 |
| Livermore-2 | 87 | 92 | 44 | 100 |
| Livermore-3 | 66 | 100 | 100 | 100 |
| Livermore-4 | 76 | 100 | 100 | 100 |
| Livermore-5 | 0 | 0 | 0 | 4 |
| Livermore-6 | 97 | 4 | 64 | 88 |
| Livermore-7 | 0 | 0 | 0 | 0 |
| Livermore-8 | 0 | 0 | 0 | 0 |
| Livermore-9 | 0 | 0 | 12 | 24 |
| Livermore-10 | 0 | 0 | 0 | 24 |
| Livermore-11 | 17 | 100 | 92 | 100 |
| Livermore-12 | 61 | 100 | 100 | 100 |
| Livermore-13 | 55 | 96 | 84 | 100 |
| Livermore-14 | 0 | 96 | 100 | 100 |
| Livermore-15 | 0 | 92 | 96 | 100 |
| Livermore-16 | 4 | 92 | 12 | 92 |
| Livermore-17 | 0 | 64 | 4 | 68 |
| Livermore-18 | 0 | 32 | 0 | 56 |
| Livermore-19 | 100 | 100 | 100 | 100 |
| Livermore-20 | 98 | 100 | 100 | 100 |
| Livermore-21 | 2 | 12 | 64 | 24 |
| Livermore-22 | 3 | 0 | 68 | 84 |
| **Livermore average** | 30.41 | 58.18 | 56.36 | **71.09** |
| **All average** | 45.19 | 63.57 | 64.11 | **74.92** |

Table 6: Single-core runtimes of our algorithm on the Nguyen benchmark problem set.

| Benchmark | Runtime (sec) |
|---|---|
| Nguyen-1 | 27.05 |
| Nguyen-2 | 59.79 |
| Nguyen-3 | 151.06 |
| Nguyen-4 | 268.88 |
| Nguyen-5 | 501.65 |
| Nguyen-6 | 43.96 |
| Nguyen-7 | 752.32 |
| Nguyen-8 | 123.21 |
| Nguyen-9 | 31.17 |
| Nguyen-10 | 103.72 |
| Nguyen-11 | 66.50 |
| Nguyen-12* | 1057.11 |
| Average | 265.54 |

Table 7: Comparison of mean root-mean-square error (RMSE) for our method to literature-reported values from DSR [Petersen et al., 2021] and Bayesian symbolic regression (BSR) [Jin et al., 2019] on the Jin benchmark problem set. Note that these benchmarks include floating-point constant values that are optimized as part of the reward function computation, as in Petersen et al. [2021]. Table 10 shows the formulas for these benchmarks.

| | Mean RMSE | | | |
|---|---|---|---|---|
| | Ours | DSR | BSR | Recovered by Ours |
| Jin-1 | **0** | 0.46 | 2.04 | Yes |
| Jin-2 | **0** | **0** | 6.84 | Yes |
| Jin-3 | **0** | 0.00052 | 0.21 | Yes |
| Jin-4 | **0** | 0.00014 | 0.16 | Yes |
| Jin-5 | **0** | **0** | 0.66 | Yes |
| Jin-6 | **0** | 2.23 | 4.63 | Yes |
| Average | **0** | 0.45 | 2.42 | |

Table 8: Comparison of median root-mean-square error (RMSE) for our method to literature-reported values from DSR [Petersen et al., 2021] and *neat* genetic programming (Neat-GP) [Trujillo et al., 2016] on the Neat benchmark problem set. Note that Neat-6, Neat-7, and Neat-8 are not fully recoverable given the function set prescribed by the benchmark. Table 10 shows the formulas for these benchmarks.

| | Median RMSE | | | |
|---|---|---|---|---|
| | Ours | DSR | Neat-GP | Recovered by Ours |
| Neat-1 | **0** | **0** | 0.0779 | Yes |
| Neat-2 | **0** | **0** | 0.0576 | Yes |
| Neat-3 | **0** | 0.0041 | 0.0065 | Yes |
| Neat-4 | **0** | 0.0189 | 0.0253 | Yes |
| Neat-5 | **0** | **0** | 0.0023 | Yes |
| Neat-6 | $\mathbf{6.1 \times 10^{-6}}$ | 0.2378 | 0.2855 | — |
| Neat-7 | **1.0028** | 1.0606 | 1.0541 | — |
| Neat-8 | **0.0228** | 0.1076 | 0.1498 | — |
| Neat-9 | **0** | 0.1511 | 0.1202 | Yes |
| Average | **0.1139** | 0.1756 | 0.1977 | |

Table 9: Benchmark symbolic regression problem specifications. Input variables are denoted by $x$ and/or $y$. $U(a, b, c)$ denotes $c$ random points uniformly sampled between $a$ and $b$ for each input variable; training and test datasets use different random seeds. $E(a, b, c)$ denotes $c$ points evenly spaced between $a$ and $b$ for each input variable; training and test datasets use the same points. All benchmark problems use the following set of allowable tokens: $\{+, -, \times, \div, \sin, \cos, \exp, \log, x, y\}$ ($y$ is excluded for single-dimensional datasets).

| Name | Expression | Dataset |
|------|-----------|---------|
| Nguyen-1 | $x^3 + x^2 + x$ | $U(-1, 1, 20)$ |
| Nguyen-2 | $x^4 + x^3 + x^2 + x$ | $U(-1, 1, 20)$ |
| Nguyen-3 | $x^5 + x^4 + x^3 + x^2 + x$ | $U(-1, 1, 20)$ |
| Nguyen-4 | $x^6 + x^5 + x^4 + x^3 + x^2 + x$ | $U(-1, 1, 20)$ |
| Nguyen-5 | $\sin(x^2)\cos(x) - 1$ | $U(-1, 1, 20)$ |
| Nguyen-6 | $\sin(x) + \sin(x + x^2)$ | $U(-1, 1, 20)$ |
| Nguyen-7 | $\log(x + 1) + \log(x^2 + 1)$ | $U(0, 2, 20)$ |
| Nguyen-8 | $\sqrt{x}$ | $U(0, 4, 20)$ |
| Nguyen-9 | $\sin(x) + \sin(y^2)$ | $U(0, 1, 20)$ |
| Nguyen-10 | $2\sin(x)\cos(y)$ | $U(0, 1, 20)$ |
| Nguyen-11 | $x^y$ | $U(0, 1, 20)$ |
| Nguyen-12 | $x^4 - x^3 + \frac{1}{2}y^2 - y$ | $U(0, 1, 20)$ |
| Nguyen-12$\star$ | $x^4 - x^3 + \frac{1}{2}y^2 - y$ | $U(0, 10, 20)$ |
| R-1 | $\frac{(x+1)^3}{x^2-x+1}$ | $E(-1, 1, 20)$ |
| R-2 | $\frac{x^5-3x^3+1}{x^2+1}$ | $E(-1, 1, 20)$ |
| R-3 | $\frac{x^6+x^5}{x^4+x^3+x^2+x+1}$ | $E(-1, 1, 20)$ |
| R-1$\star$ | $\frac{(x+1)^3}{x^2-x+1}$ | $E(-10, 10, 20)$ |
| R-2$\star$ | $\frac{x^5-3x^3+1}{x^2+1}$ | $E(-10, 10, 20)$ |
| R-3$\star$ | $\frac{x^6+x^5}{x^4+x^3+x^2+x+1}$ | $E(-10, 10, 20)$ |
| Livermore-1 | $\frac{1}{3} + x + \sin(x^2)$ | $U(-10, 10, 1000)$ |
| Livermore-2 | $\sin(x^2)\cos(x) - 2$ | $U(-1, 1, 20)$ |
| Livermore-3 | $\sin(x^3)\cos(x^2) - 1$ | $U(-1, 1, 20)$ |
| Livermore-4 | $\log(x + 1) + \log(x^2 + 1) + \log(x)$ | $U(0, 2, 20)$ |
| Livermore-5 | $x^4 - x^3 + x^2 - y$ | $U(0, 1, 20)$ |
| Livermore-6 | $4x^4 + 3x^3 + 2x^2 + x$ | $U(-1, 1, 20)$ |
| Livermore-7 | $\sinh(x)$ | $U(-1, 1, 20)$ |
| Livermore-8 | $\cosh(x)$ | $U(-1, 1, 20)$ |
| Livermore-9 | $x^9 + x^8 + x^7 + x^6 + x^5 + x^4 + x^3 + x^2 + x$ | $U(-1, 1, 20)$ |
| Livermore-10 | $6\sin(x)\cos(y)$ | $U(0, 1, 20)$ |
| Livermore-11 | $\frac{x^2 y^2}{x+y}$ | $U(-1, 1, 50)$ |
| Livermore-12 | $\frac{x^5}{y^3}$ | $U(-1, 1, 50)$ |
| Livermore-13 | $x^{\frac{1}{3}}$ | $U(0, 4, 20)$ |
| Livermore-14 | $x^3 + x^2 + x + \sin(x) + \sin(x^2)$ | $U(-1, 1, 20)$ |
| Livermore-15 | $x^{\frac{1}{5}}$ | $U(0, 4, 20)$ |
| Livermore-16 | $x^{\frac{2}{5}}$ | $U(0, 4, 20)$ |
| Livermore-17 | $4\sin(x)\cos(y)$ | $U(0, 1, 20)$ |
| Livermore-18 | $\sin(x^2)\cos(x) - 5$ | $U(-1, 1, 20)$ |
| Livermore-19 | $x^5 + x^4 + x^2 + x$ | $U(-1, 1, 20)$ |
| Livermore-20 | $\exp(-x^2)$ | $U(-1, 1, 20)$ |
| Livermore-21 | $x^8 + x^7 + x^6 + x^5 + x^4 + x^3 + x^2 + x$ | $U(-1, 1, 20)$ |
| Livermore-22 | $\exp(-0.5x^2)$ | $U(-1, 1, 20)$ |

Table 10: Benchmark symbolic regression problems that include unknown constants. Input variables are denoted by $x$ and/or $y$. $U(a, b, c)$ denotes $c$ random points uniformly sampled between $a$ and $b$ for each input variable; training and test datasets use different random seeds. $E(a, b, c)$ denotes $c$ points evenly spaced between $a$ and $b$ for each input variable; training and test datasets use the same points (except Neat-6, which uses $E(1, 120, 120)$ as test data, and the Jin tests, which use $U(-3, 3, 30)$ as test data). To simplify notation, libraries are defined relative to a "base" library $\mathcal{L}_0 = \{+, -, \times, \div, \sin, \cos, \exp, \log, x\}$. Placeholder operands are denoted by $\bullet$, e.g. $\bullet^2$ corresponds to the square operator.

| Name | Expression | Dataset | Library |
|------|-----------|---------|---------|
| Jin-1 | $2.5x^4 - 1.3x^3 + 0.5y^2 - 1.7y$ | $U(-3, 3, 100)$ | $\mathcal{L}_0 - \{\log\} \cup \{\bullet^2, \bullet^3, y, \text{const}\}$ |
| Jin-2 | $8.0x^2 + 8.0y^3 - 15.0$ | $U(-3, 3, 100)$ | $\mathcal{L}_0 - \{\log\} \cup \{\bullet^2, \bullet^3, y, \text{const}\}$ |
| Jin-3 | $0.2x^3 + 0.5y^3 - 1.2y - 0.5x$ | $U(-3, 3, 100)$ | $\mathcal{L}_0 - \{\log\} \cup \{\bullet^2, \bullet^3, y, \text{const}\}$ |
| Jin-4 | $1.5\exp(x) + 5.0\cos(y)$ | $U(-3, 3, 100)$ | $\mathcal{L}_0 - \{\log\} \cup \{\bullet^2, \bullet^3, y, \text{const}\}$ |
| Jin-5 | $6.0\sin(x)\cos(y)$ | $U(-3, 3, 100)$ | $\mathcal{L}_0 - \{\log\} \cup \{\bullet^2, \bullet^3, y, \text{const}\}$ |
| Jin-6 | $1.35xy + 5.5\sin((x - 1.0)(y - 1.0))$ | $U(-3, 3, 100)$ | $\mathcal{L}_0 - \{\log\} \cup \{\bullet^2, \bullet^3, y, \text{const}\}$ |
| Neat-1 | $x^4 + x^3 + x^2 + x$ | $U(-1, 1, 20)$ | $\mathcal{L}_0 \cup \{1\}$ |
| Neat-2 | $x^5 + x^4 + x^3 + x^2 + x$ | $U(-1, 1, 20)$ | $\mathcal{L}_0 \cup \{1\}$ |
| Neat-3 | $\sin(x^2)\cos(x) - 1$ | $U(-1, 1, 20)$ | $\mathcal{L}_0 \cup \{1\}$ |
| Neat-4 | $\log(x + 1) + \log(x^2 + 1)$ | $U(0, 2, 20)$ | $\mathcal{L}_0 \cup \{1\}$ |
| Neat-5 | $2\sin(x)\cos(y)$ | $U(-1, 1, 100)$ | $\mathcal{L}_0 \cup \{y\}$ |
| Neat-6 | $\sum_{k=1}^{x} \frac{1}{k}$ | $E(1, 50, 50)$ | $\{+, \times, \div, \bullet^{-1}, -\bullet, \sqrt{\bullet}, x\}$ |
| Neat-7 | $2 - 2.1\cos(9.8x)\sin(1.3y)$ | $E(-50, 50, 10^5)$ | $\mathcal{L}_0 \cup \{\tan, \tanh, \bullet^2, \bullet^3, \sqrt{\bullet}, y\}$ |
| Neat-8 | $\frac{e^{-(x-1)^2}}{1.2 + (y - 2.5)^2}$ | $U(0.3, 4, 100)$ | $\{+, -, \times, \div, \exp, e^{-\bullet}, \bullet^2, x, y\}$ |
| Neat-9 | $\frac{1}{1 + x^{-4}} + \frac{1}{1 + y^{-4}}$ | $E(-5, 5, 21)$ | $\mathcal{L}_0 \cup \{y\}$ |

Table 11: Hyperparameter values for all experiments, unless otherwise noted. If an applicable hyperparameter value differs from one of the baseline methods, it is noted.

| Hyperparameter | Symbol | Value | Comment |
|---|---|---|---|
| **Shared Parameters** | | | |
| Batch/population size | $N$ | 500 | Petersen et al. [2021]: 1000 |
| | | | Fortin et al. [2012]: 300 |
| Minimum expression length | – | 4 | – |
| Maximum expression length | – | 30 | – |
| Maximum expressions | – | 2,000,000 | – |
| Maximum constants | – | $\infty$ | Petersen et al. [2021]: 3 |
| | | | (only used for Jin benchmarks) |
| Reward/fitness function | $R$ | Inverse NRMSE | – |
| **RNN Parameters** | | | |
| Optimizer | – | Adam | – |
| RNN cell type | – | LSTM | – |
| RNN cell layers | – | 1 | – |
| RNN cell size | – | 32 | – |
| Training method | – | PQT | – |
| PQT queue size | – | 10 | – |
| Sample selection size | $M$ | 1 | – |
| Learning rate | $\alpha$ | 0.0025 | Petersen et al. [2021]: 0.0005 |
| Entropy weight | – | 0.005 | – |
| **GP Parameters** | | | |
| Generations per iteration | $S$ | 25 | – |
| Crossover operator | – | One Point | – |
| Crossover probability | $P_c$ | 0.5 | – |
| Mutation operator | – | Multiple | Fortin et al. [2012]: Uniform |
| Mutation probability | $P_m$ | 0.5 | Fortin et al. [2012]: 0.1 |
| Selection operator | – | Tournament | – |
| Tournament size | $k$ | 5 | Fortin et al. [2012]: 3 |
| Mutate tree maximum | – | 3 | Fortin et al. [2012]: 2 |

Table 12: Additional ablations when using original hyperparameter values from Petersen et al. [2021] and/or Fortin et al. [2012] rather than the values in Table 11.

| | Recovery rate (%) | | | |
|---|---|---|---|---|
| | All | Nguyen | R | Livermore |
| Mutation probability $P_m = 0.1$ | 75.24 | 90.33 | 32.00 | 72.91 |
| Baseline (Table 11 values) | 74.92 | 92.33 | 33.33 | 71.09 |
| Tournament size $k = 3$ | 73.62 | 92.00 | 29.33 | 69.64 |
| Batch size $N = 1000$ | 72.11 | 93.33 | 25.33 | 66.91 |
| RNN learning rate $\alpha = 0.0005$ | 71.03 | 91.67 | 22.67 | 66.36 |
| 95% confidence interval | ±1.54 | ±1.76 | ±2.81 | ±1.32 |

Table 13: Recovery rates when using the soft length prior (SLP) and hierarchical entropy regularizer (HER) introduced in Landajuela et al. [2021a], and increasing maximum length from 30 to 100. These results are post-hoc, as Landajuela et al. [2021a] was performed concurrently.

| | Recovery rate (%) | |
|---|---|---|
| | Ours | Ours + SLP/HER |
| Nguyen-1 | 100 | 100 |
| Nguyen-2 | 100 | 100 |
| Nguyen-3 | 100 | 100 |
| Nguyen-4 | 100 | 100 |
| Nguyen-5 | 100 | 100 |
| Nguyen-6 | 100 | 100 |
| Nguyen-7 | 96 | 100 |
| Nguyen-8 | 100 | 100 |
| Nguyen-9 | 100 | 100 |
| Nguyen-10 | 100 | 100 |
| Nguyen-11 | 100 | 100 |
| Nguyen-12* | 12 | 4 |
| **Nguyen average** | **92.33** | 92.00 |
| R-1* | 4 | 100 |
| R-2* | 4 | 100 |
| R-3* | 92 | 96 |
| **R average** | 33.33 | **98.67** |
| Livermore-1 | 100 | 100 |
| Livermore-2 | 100 | 100 |
| Livermore-3 | 100 | 100 |
| Livermore-4 | 100 | 100 |
| Livermore-5 | 4 | 40 |
| Livermore-6 | 88 | 100 |
| Livermore-7 | 0 | 4 |
| Livermore-8 | 0 | 0 |
| Livermore-9 | 24 | 88 |
| Livermore-10 | 24 | 8 |
| Livermore-11 | 100 | 100 |
| Livermore-12 | 100 | 100 |
| Livermore-13 | 100 | 100 |
| Livermore-14 | 100 | 100 |
| Livermore-15 | 100 | 100 |
| Livermore-16 | 92 | 100 |
| Livermore-17 | 68 | 36 |
| Livermore-18 | 56 | 48 |
| Livermore-19 | 100 | 100 |
| Livermore-20 | 100 | 100 |
| Livermore-21 | 24 | 88 |
| Livermore-22 | 84 | 92 |
| **Livermore average** | 71.09 | **77.45** |
| **All average** | 74.92 | **83.89** |