# OpenReview forum: "Symbolic Regression via Deep Reinforcement Learning Enhanced Genetic Programming Seeding"
_NeurIPS.cc/2021/Conference — NeurIPS 2021 Poster_

### Official Review · Reviewer_DKkf · 2021-06-27

**Rating:** 6
**Confidence:** 3

**Summary:**

The manuscript proposes a method to solve symbolic regression problems.  Recurrent  neural networks (RNNs)  are trained using policy-gradient reinforcement learning to find initial populations for genetic programming (GP).



**Limitations And Societal Impact:**

In my evaluation, no additional discussion of potential negative societal impact of the presented work is necessary.

**Main Review:**

The overall approach is mainly a combination of existing methods. Thus, I regard the originality as limited.
I did not find obvious flaws or misleading statements in the manuscript.
It is interesting that GP alone performs so well (table 2).

Some part were not clear to me:
lines 193-195: Are the individuals which violate the constraints removed (i.e., get lethal fitness) or repaired (e.g., ln(e^x) is simply replaced by x)?

lines 54-59:  The authors could elaborate what they exactly refer to with "out-of-distribution", "off-policy", and "on-policy" in this context. What are the two policies? Would the authors regard the RNN to be on-policy and, if yes, why is this a problem?

lines 231-234: I am not fully sure that I understand the fundamental difference between PQT and RSPG: PQT takes the best from T_rl and T_gp combines while RSPG only removes the worst solutions from T_rl?

line 303: The entropy term comes out of the blue - where is it used?

Literature embedding:

lines 87-89: the idea of combining GP and gradient-based optimization of the numerical values goes back to:
C. Igel. M. Kreutz. Using Fitness Distributions to Improve the Evolution of Learning Structures. Congress on Evolutionary Computation (CEC 99), pp. 1902-1909, IEEE Press, 1999 [see section 5]


Minor comments:

line 27: ", is" -> "is"

line 48: "the problem" -> "the general problem" ?

line 419: first name abbreviations



**Time Spent Reviewing:**

2

---

> ### Author Response · Authors · 2021-08-10
> **Response to Reviewer DKkf**
>
> **Combining existing methods.** Please see our top-level comment regarding the novelty of our approach.
>
> **Mechanics of constraint violation.** We clarify what happens when an individual violates a constraint as follows. If a genetic operation would result in an individual that violates any constraint, that genetic operation is instead reverted. That is, the child expression instead becomes a copy of the parent expression. This procedure ensures that all individuals satisfy all constraints for all generations. Regarding the suggested alternatives: (1) We cannot simply assign a lethal fitness to individuals with violated constraints, because such individuals passed back to the RNN would have zero likelihood; thus, the gradient of the objective would no longer be well-defined. (2) While some violations could simply be corrected (e.g. $\log(\exp(x)) \rightarrow x$ as suggested), this is not the case in general, e.g. for trigonometric operators with trigonometric operator descendants.
>
> **Off-policy vs on-policy clarification.** The policy is the RNN. Thus, we consider samples from the RNN to be on-policy, while individuals returned from GP are off-policy. "Out-of-distribution" thus refers to a situation that may occur if GP generates a sample (via evolutionary operators) that has a relatively low likelihood under the RNN.
>
> **PQT vs RSPG.** The reviewer correctly states that one difference between PQT and RSPG is that PQT takes the best from $T_{rl}$ and $T_{gp}$, whereas RSPG only takes the best from $T_{rl}$. The reason for this is that PQT is a greedy heuristic method that aims to reproduce the best expressions ever seen; in contrast, RSPG is a theory-driven approach that maximizes the quantile-conditioned expected reward; the theory *prescribes* that filtering the best expression occurs each batch and only for on-policy samples (i.e. samples from the RNN). This fundamental difference leads to two additional algorithmic differences between PQT and RSPG: (1) PQT maintains a *persistent* priority queue that trains on the best expressions *ever seen* during training, whereas RSPG trains on the best expressions sampled *each batch*. (2) PQT trains via a cross-entropy objective (i.e. supervised learning), whereas RSPG trains via a policy gradient objective (i.e. reinforcement learning). We will clarify these subtle differences in our revised text.
>
> **Entropy term.** Inclusion of an entropy regularizer in the loss function is standard practice for neural-guided search. This is based on the *maximum entropy reinforcement learning* framework [Haarnoja et al., 2018]. This term provides a bonus to the loss function proportional to the entropy of the sampled expressions.
>
> **Prior work combining GP with gradient-based methods.** Thank you for pointing out this earlier reference regarding combining GP with gradient-based methods. We have added this citation to our manuscript.
>
> **Typos.** Thank you for pointing out typos; we have fixed them in the revised text.
>
> References:
>
> Haarnoja, T.,  Zhou, A., Abbeel,  P., and Levine, S.(2018). Soft actor-critic: Off-policy maximum entropy deep reinforcement learning with a stochastic actor. ICML 2018.

---

### Official Review · Reviewer_ELHf · 2021-07-16

**Rating:** 5
**Confidence:** 2

**Summary:**

This paper proposes to combine the policy gradient method with the genetic algorithm method to solve the symbolic regression problem. Basically, an RNN trained using policy gradient computes candidate equations, and these equations are served as "seeds" for the genetic algorithm. The output of the genetic algorithm is used to train the RNN through policy gradient. Then this cycle can forward. The authors test their method on various datasets and achieve the best performance among all the baselines.

**Limitations And Societal Impact:**

N/A.

**Main Review:**

Originality:  This work is somehow a novel combination of "Deep symbolic regression: Recovering mathematical expressions from data via risk-seeking policy gradients Petersen et al. 2021" and " Guiding deep molecular optimization with genetic exploration Ahn et al. 2020". The related work section is sound.

Quality: The idea of combining reinforcement learning (RL) and genetic algorithm is quite simple. Use the results computed by the RL policy to guide the genetic algorithm, and the improved results from the genetic algorithm can be used to improve the RL policy. Also, several methods to improve this cycle are studied and explained.  In the merit of simple, the performance looks good.

Clarity: It would be better if the authors can improve the writing. It is a little bit hard for me to follow the paper. Too many details are mentioned in the main text and hard to capture the high-level idea. Also, for the experiment section, it would be better if the author can mark your results at least in bold.

Significance: The program this paper study is significant.

Questions:

1. What's the difference between DSR in Tabel1 and DSR (RL only) in Tabel 2? If there are differences, why use two different baselines? And why not run all the baselines on all the datasets?

2. In terms of the fairness of comparison. I think you run 100 times for each method seems not fair. It would be better if you set a time budget and let all the methods run under the budget.

3. It seems that Nguyen-10 and 11 are more complicated than 12. But all the methods fail on 12. Do you have some intuitions?

**Time Spent Reviewing:**

3

---

> ### Author Response · Authors · 2021-08-10
> **Response to Reviewer ELHf**
>
> **Clarity.** We are committed to improving the clarity in our final manuscript. In particular, we will ensure the high-level idea is captured before describing methodological details. We will also demarcate our algorithm's results using boldface text, as suggested.
>
> **Table 2 labels.** "DSR" in Table 1 is the same algorithm as "DSR (RL only)" in Table 2. We will clarify this in our manuscript.
>
> **Choice of baselines.** Since Petersen et al. 2021 was previous SOTA, the goal of Table 1 was to provide a direct, "head-to-head" comparison to results reported therein, using the same set of baselines and benchmark datasets (Nguyen-1 to Nguyen-12). After this study, we dropped VPG and vanilla GP (i.e. GP without random restarts) because their performance is very poor, and Eureqa because our institution does not have access to this commercial software license.
>
> The remainder of our empirical studies are on multiple benchmark suites (Tables 2 and 3). For these studies, we focus on the most competitive baselines and many ablations of our algorithm.
>
> Thus, regarding the question of why not use all baselines for all experiments, the only baselines not included for all benchmarks were either not competitive (VPG, GP) or required a software license (Eureqa).
>
> **Comparison fairness.** We first clarify: for each algorithm, each experiment has a budget of 2M expression evaluations; we repeat these experiments 100 times for each algorithm.
>
> We maintain that a budget based on the number of expression evaluations (2M) makes for much fairer comparisons than a runtime-based budget because it removes the conflating factor of which algorithm's codebase was more optimized for performance. Further, we note that for all algorithms considered, the computational bottleneck is evaluating the reward/fitness of expressions. The gradient updates and/or genetic operations are computationally cheap relative to this. Further, since we allow all algorithms to stop early if/when a correct solution is found, the dominating factor determining total runtime is the recovery rate. Thus, we believe a runtime-based budget would give an unfair advantage to *our* algorithm (DREGS), which yields the highest recovery rate.
>
> To demonstrate, the mean runtime on the Nguyen benchmark suite is 265.5 seconds for DREGS and 615.2 seconds for DSR, a 2.3-fold improvement. This is largely due to DREGS yielding the highest recovery rate, which allows it to stop early. For Nguyen-12 (which is never recovered) specifically, we observe very similar mean runtimes across algorithms: 1057.1 seconds for DREGS and 1055.4 seconds for DSR. This supports our claim that runtime is dominated by the number of expression evaluations, and algorithmic differences alter runtime only very slightly. We hope this allays all concerns regarding fairness of comparison based on computational budget.

---

### Official Review · Reviewer_Ug8S · 2021-07-17

**Rating:** 9
**Confidence:** 5

**Summary:**

The paper describes a method to enhance the performance of Genetic Programming (GP) for symbolic regression by iteratively enhancing initial populations for 'random restart' GP implementations. The method works through a generative recurrent network (RNN) and an Reinforcement Learning (RL) approach, in which the RNN gets rewarded when outputs a set of initial math expressions that help the GP to achieve better solutions after each run. The GP and the RL algorithm work cooperatively in an iterative fashion where for every GP run (25 cycles) the RL rewards the RNN. Authors provide an extensive battery of tests in several benchmarking datasets, as well as proposing their own set of problems. Their method achieves better results than all other methods they compare against, including a vanilla GP.

**Limitations And Societal Impact:**

Authors do not provide any discussion regarding potential societal impacts of their work; nevertheless, considering it is a highly theoretical work, a purely algorithmic contribution, I do not consider such analysis a requirement for this paper.

**Main Review:**

The paper is very well written and structured. Language is clear at all times; math is not overused.

The method is very well described and presented. I really like the simplicity behind it, and the authors do a good work explaining the motivations and intuitions behind it.

The contribution, however, can be arguably minor, because they are taking two solid methods and just gluing them together; nevertheless, it works, so I'm guessing authors will have to defend the relevance of their contribution to the other reviewers.

I personally consider it worthy for acceptance at NeurIPS; there are, however, some details that I consider that need attention, before I can give a higher evaluation for the paper:

- I reviewed closely your experimental design, and I noticed that you use extremely small sample points sets (10~20 points); I wonder how results change if you provide more sample points, let's say, in the range of hundreds or thousands, to the different methods tested.

- If I understood correctly, your 'primitives' (as called in GP literature) is {+,−,×,÷,sin,cos,exp,log,x} (from your appendix). So, there are certain problems, such as Nguyen-8 and Nguyen-11 where the GP has no chance to find the correct expression, because it would be simply too difficult, or even impossible, to build it given those primitives.... why or how did you expect the GP to find those expression?? For example, express to me x^y using only the set of functions mentioned above. But more importantly, how did your method manage to find such functions?

- The most important element that I find missing in your draft, is a bit of discussion on how or why you think your method works, at a low level analysis. I do understand the perks and capabilities of RL, but my question is... what exactly is doing the RL that enhances that much a vanilla GP. My bet is that is pruning the primitives set, such that the GP has a smaller and better defined search space, but I'd really like to hear your thoughts on this.

Notice how your method could help immensely to understand on why GP sometimes work, sometimes doesn't, i.e. what makes a problem difficult for GP, which has been the million dollar question in GP research, but you need to provide a bit of discussion in such regard.

[UPDATED RECOMMENDATION]
Authors properly addressed all my questions and concerns. I've no doubt on the effectiveness and relevance of their method. The paper is well written, and my main objection (lack of discussion on how the method works at a low level) has been attended. Therefore, I see no real reason for this work for not being accepted.

**Time Spent Reviewing:**

7

---

> ### Author Response · Authors · 2021-08-10
> **Response to Reviewer Ug8S**
>
> **Please see our top-level response.** Regarding "gluing" together two solid methods, and intuition on why the method works, please see our top-level comment which addresses these two topics.
>
> **Number of sample points.** For all algorithms, increasing the number of sample points serves to *smooth* the reward/fitness function. Thus, it is not expected to favor any one algorithm over another. For this reason, we did not repeat experiments for different dataset sizes. Further, the dataset size is part of the problem definition of the benchmark suite as defined by Nguyen and R, not selected by us. (For our Anonymous benchmark suites, we defined similar dataset sizes as existing benchmark suites.) Finally, we note that symbolic regression excels in the low-data setting, as they are less prone to overfitting than, for example, MLP models; this is why benchmark suites tend to use small datasets.
>
> **Recovering Nguyen-8 and Nguyen-11.** The primitives listed (as defined by the benchmark authors) are correct; however, one can express both Nguyen-8 and Nguyen-11 analytically using these primitives. Specifically, one can express $x^y$ (Nguyen-11) via the identity $x^y = \exp ( y \log x )$, which holds for $x \in \mathbb{R}^{+}$. This can be represented via the example pre-order traversal $[ \exp, \times, y, \log, x ]$. Similarly, one can express $\sqrt{x}$ (Nguyen-8) via the identity $\sqrt{x} = \exp\left(\frac{1}{2} \log x\right)$, which holds for $x \in \mathbb{R}^+$. This can be represented via the example pre-order traversal $[ \exp, \times, \log, x,  \div, x, +, x, x ]$. Note that integers are not included in the primitives for the Nguyen benchmarks, so the factor of $\frac{1}{2}$ must be expressed through an identity like $\frac{1}{2} = \frac{x}{x+x}$, which holds for $x \neq 0$. For both Nguyen-8 and Nguyen-11, $x > 0$ for all data points, so all identities hold. We recognize that these exact solutions are non-obvious and thus we will make note of them in our revised text.
>
> **GP problem difficulty.** We strongly agree that assessing problem difficulty is a grand challenge, not just for GP, but also for discrete optimization more broadly. We also agree that hybrid methods like the one we propose can be useful for such analysis. For example, one can compare which problems lead to failure for GP-based approaches versus RNN-based approaches. Knowledge of these differences could also be useful in tuning hybrid approaches, e.g. placing more or less of the learning on RNN vs GP. As this work is focused on developing a hybridization framework and comparing against performance, we save such post-mortem analysis for future work.

---

> > ### Comment · Reviewer_Ug8S · 2021-08-18
> > **Thanks for response; upgrading my recommendation score**
> >
> > First of all, I'd like to thank authors for properly addressing my concerns and questions regarding their work.
> > I consider that this work could be disruptive for the GP framework, thus I really would like to see it published. I'll update my recommendation score accordingly.

---

### Official Review · Reviewer_wnag · 2021-07-19

**Rating:** 6
**Confidence:** 2

**Summary:**

This paper proposes a hybrid method of policy gradient-based reinforcement learning and genetic programming for symbolic regression. The solutions provided by the policy-gradient-based deep reinforcement learner (DRL) are used as the population in genetic programming. The experimental result using benchmark functions for symbolic regression shows that the proposed method is superior to the existing symbolic regression methods in most cases.

**Limitations And Societal Impact:**

The reviewer thinks that the limitation is fairly stated.

**Main Review:**

The hybrid method of a policy gradient method and genetic programming (GP) is interesting. However, the combination method proposed in this paper seems to be simple. In the proposed method, the sampled solutions from a deep neural network are used as the initial population of GP iterations. In addition, the generated solutions by GP are injected into the samples used for updating the neural network weights via policy gradient.

The samples generated by GP iterations are used for updating neural network weights by policy gradient update. The policy gradient formula shown in section 3 is the Monte-Carlo approximation of the true gradient under the current policy. If we inject the samples generated by GP to calculate the policy gradient, the estimation of the policy gradient is biased. It should be carefully noted the justification of the proposed method.

The reviewer cannot understand the reason for the performance improvement by the proposed method. How does genetic programming contribute to improving the recovery rate? The authors should elaborate on the reason for the performance improvement.

In [Petersen et al., 2021], other benchmark symbolic regression problems are also examined, such as Jin-X and Neat-X. How about the performance of the proposed method on these benchmarks?

Does "GP-Meld" mean "DREGS" in Table 4?


-----
[Update after rebuttal]
The authors reply to my request properly. I agree with the clear performance improvement by the proposed method as the strength of this paper. SO, I would raise my score. However, the novelty is limited because the proposed method is a simple heuristic combination of the existing methods.

**Time Spent Reviewing:**

3

---

> ### Author Response · Authors · 2021-08-10
> **Response to Reviewer wnag**
>
> **Please see our top-level response.** Regarding the simplicity of our proposed approach, and discussion on how GP improves recovery rate, please see our top-level comment which addresses these two topics.
>
> **GP samples.** The reviewer is correct that samples generated by GP bias the policy gradient estimate. We acknowledge and discuss this fact in lines 51-59 of the manuscript. In this work, we are deliberately relaxing the strict RL objective of maximizing expected returns to explore the hybridization of GP with *several* RL-inspired methods for neural-guided search, for example PQT, which itself is not optimizing expected return. Thus, we view our resulting hybridized methods as more closely related to the cross-entropy method [Rubinstein and Kroese, 2013], in which an elite set is provided by an orthogonal sampling process (in our case, defined by the GP). The connection between the cross-entropy method and policy gradients is well established and understood (see [Schulman, 2016]). On the other hand, the method can also be regarded as an expert iteration algorithm [Anthony et al., 2017], where the RNN acts as an apprentice and the GP as an expert providing expert guidance.
>
> Lastly, we note that the strong empirical performance of our algorithm when using RL objectives (RSPG and VPG) suggests that the bias in the gradient estimate is not detrimental to performance.
>
> **Neat and Jin benchmarks.** Yes, Petersen et al. 2021 includes additional baselines in their supplement. We are currently running these experiments and will report back when finished. These runs take considerably longer as there is a constant optimizer used for each reward computation.
>
> **Typo.** Yes, "GP-Meld" should be "DREGS" (our algorithm) in Table 4; we revised this in the manuscript. Thank you for pointing this out.
>
> References:
>
> Anthony, T., Tian, Z., and Barber, D. (2017). Thinking fast and slow  with  deep learning and tree search.
>
> Rubinstein, R. Y. and  Kroese, D. P. (2013). The cross-entropy method: a unified approach to combinatorial optimization, Monte-Carlo simulation and machine learning. Springer Science \& Business Media.
>
> Schulman, J. (2016). Optimizing expectations: From deep reinforcement learning to stochastic computation graphs. PhD thesis, UC Berkeley.

---

> > ### Comment · Reviewer_wnag · 2021-08-18
> > **Thank you for the response**
> >
> > Thank you for reporting the additional experimental result. I understand the superiority of the proposed method in the wide range of benchmarks.
> >
> > Also, thank you for the explanation regarding the biased samples by GP. I think that justifying the usage of such biased samples is important for theoretical motivation. I understand that it relates to the cross-entropy method and expert iteration algorithm.

---

> ### Author Response · Authors · 2021-08-11
> **Jin and Neat benchmarks**
>
> **Jin and Neat experiments.** Our experiments completed on the Jin and Neat benchmark suites. We carefully mimicking the experimental setup described in Petersen et al., 2021 (mean RMSE per benchmark for Jin and median RMSE per benchmark for Neat). In the tables below, we compare the performance of our algorithm (DREGS) with the results published in Petersen et al., 2021. DREGS outperforms DSR (or ties, in the cases where DSR also achieves zero RMSE) for all benchmarks.
>
> | | DREGS (ours) | DSR (Petersen et al., 2021) |
> |-|:-:|:-:|
> | Jin-1 | **0.36** | 0.46
> | Jin-2 | **0.00** | **0.00**
> | Jin-3 | **0.00** | $5.20 \times 10^{-4}$
> | Jin-4 | **0.00** | $1.36 \times 10^{-4}$
> | Jin-5 | **0.00** | **0.0**
> | Jin-6 | **0.23** | 2.23
> | Mean | **0.10** | 0.45
>
> | | DREGS (ours) | DSR (Petersen et al., 2021) |
> |-|:-:|:-:|
> | Neat-1 | **0.00** | **0.00**
> | Neat-2 | **0.00** | **0.00**
> | Neat-3 | **0.00** | $4.10 \times 10^{-3}$
> | Neat-4 | **0.00** | $1.89 \times 10^{-2}$
> | Neat-5 | **0.00** | **0.0**
> | Neat-6 | $\mathbf{1.40 \times 10^{-4}}$ | 0.24
> | Neat-7 | **1.05** | 1.06
> | Neat-8 | **0.03** | 0.11
> | Neat-9 | **0.00** | 0.15
> | Mean | **0.12** | 0.18

---

### Author Response · Authors · 2021-08-10
**Comment to all reviewers**

We thank all reviewers for their time and useful feedback. We wish to address two main points shared by multiple reviewers, first regarding the simplicity of the proposed approach, and second regarding intuition as to why the approach yields such strong empirical results.

**Simplicity.** In general, we agree that our approach is mechanically simple; however, we believe the contribution of our approach is that it is novel and substantially outperforms existing SOTA baselines. Were the results only a *minor* improvement upon existing methods, we agree that the method would be less useful to other ML researchers and practitioners. Moreover, we consider the simplicity of our approach an advantage, making it more readily usable by others.

To the best of our knowledge, the only other method that uses evolutionary algorithms within an inner optimization loop of neural-guided search is GEGL [Ahn et al., 2020], which we compare to in Table 2 and demonstrate substantially higher performance. Further, we note GEGL is tied to one specific (PQT-like) algorithm for neural-guided search. In contrast, we demonstrated our approach in the context of several other neural-guided search algorithms (RSPG, VPG, and PQT). Thus, our work provides a new modular framework for hybridizing one *class* of methods (neural-guided search) with another class (evolutionary/population-based algorithms).

**Why the algorithm works.** We provide a two-sided perspective to provide intuition as to why the algorithm works so well. We first describe how the neural-guided search component (for simplicity, RNN) benefits from the evolutionary component (for simplicity, GP), and then we describe how GP benefits from the RNN.

From the RNN perspective, GP helps by providing "fresh" new samples that help escape local optima. NN-based discrete distributions often concentrate their probability mass on a relatively small portion of the search space, resulting in premature convergence to locally optimal solutions. This phenomenon is well established in the policy gradient literature. In contrast, the evolutionary operations of GP generate new individuals that may fall well outside the RNN's concentrated region of the search space. We empirically demonstrate this by comparing the log-likelihood of RNN-produced samples to the log-likelihood (under the RNN) of GP-produced samples. For example, when training DREGS on Nguyen-12, after 10 iterations the RNN produces a batch of expressions with log-likelihoods of $-29.2 \pm 7.5$, whereas the returned GP samples for that iteration have log-likelihoods of $-36.5 \pm 7.3$. This corresponds to a >1500-fold decrease in mean likelihood between the RNN and GP samples. Thus, it is extremely unlikely the RNN would have sampled these GP expressions under its current parameters. Given that DREGS substantially outperforms DSR and GP alone (as well as the "all on-policy" baseline in Table 3), we conclude that these unlikely samples are critical to achieving high performance.

From the perspective of GP, the RNN helps by *learning* good starting populations. With a good starting population, GP can easily identify excellent solutions within a small number of generations. This learning is possible because the RNN has *memory* (i.e. weights) that persist across training iterations, unlike GP which has no memory and is prone to "drift" [Doerr et al. 2020]. As one reviewer astutely noted, a particular manifestation of this learning is that the RNN learns a good distribution over tokens within the primitive set; starting populations with the right frequencies of tokens makes the GP search substantially easier. In the table below, we demonstrate this point by showing the token frequencies under the RNN at the start vs end of training on Nguyen-4 (a simple polynomial, obtainable via $x$, $+$, and $\times$ tokens only). Frequencies of required tokens increase dramatically, where unnecessary tokens drop to 1 - 3%.

| | $x$ |$+$|$-$|$\times$|$\div$|$\sin$|$\cos$|$\exp$|$\log$
| --- | ----------- |||||||||
| Start of training | 42%|9%|8%|9%|9%|4%|3%|8%|8%
| End of training | 49%|17%|1%|24%|3%|1%|1%|3%|1%

Taken together, we see that our hybrid method leverages the best of both worlds: the memory/learning capacity of the RNN and the ability of GP to easily escape local optima. Conversely, they help alleviate each other's weakness: where RNNs can get stuck in local optima, GP can provide fresh new samples; where GP's lack of memory can result in "drift," the RNN anchors it by seeding GP with informed starting populations.

References:

Ahn, S., Kim, J., Lee, H., and Shin, J. (2020). Guiding  deep  molecular  optimization with genetic exploration. NeurIPS 2020.

Doerr, B. and Zheng, W. (2020). From understanding genetic drift to a smart-restart parameter-less compact genetic algorithm. In Proceedings of the 2020 Genetic and Evolutionary Computation Conference.

---

### Decision · Program_Chairs · 2021-09-27

**Decision:**

Accept (Poster)

**Comment:**

This paper received review scores with a high variance: 5,6,6,9. The paper was actively discussed, both with the authors and in private.

Reviewer Ug8S (score: 9) championed the paper for its simplicity and strong results. Yet, the review text and the arguments given (e.g., "I see no real reason for this work for not being accepted.") do not sound like a score of 9 at NeurIPS, more like a 6 or 7).

Negative points in the reviews and the discussion were the lack of originality (the paper heuristically combines existing methods: RL in order to sample the initial population of evolutionary methods, to solve symbolic regression problems).

A brief note about the NeurIPS checklist: the authors' responses to the questions are very casual (e.g. a simple "Yes" would be preferable over "It would be a bad idea not to"), as if these points are obviously satisfied; however, the paper did *not* attach the code and as the code repo merely gave "www.anonymous.submission". That does *not* count as the code being available; it should be available for reviewers to check in an anonymized repo or attached as supplementary material.

Overall, in view of the strong results and the simplicity of the method, I am leaning towards acceptance as a poster.